# Enhancing semantic segmentation in chest X-ray images through image preprocessing: ps-KDE for pixel-wise substitution by kernel density estimation

**Yuanchen Wang, Yujie Guo, Ziqi Wang, Linzi Yu, Yujie Yan⬮, Zifan Gu⬮***

Department of Biomedical Informatics, Harvard Medical School, Boston, Massachusetts, United States of America

* zifan.gu@utsouthwestern.edu

## Abstract

**Data Availability Statement:** All radiographs are held in a public repository from the Japanese

### Background

In medical imaging, the integration of deep-learning-based semantic segmentation algorithms with preprocessing techniques can reduce the need for human annotation and advance disease classification. Among established preprocessing techniques, Contrast Limited Adaptive Histogram Equalization (CLAHE) has demonstrated efficacy in improving segmentation algorithms across various modalities, such as X-rays and CT. However, there remains a demand for improved contrast enhancement methods considering the heterogeneity of datasets and the various contrasts across different anatomic structures.

### Method

This study proposes a novel preprocessing technique, ps-KDE, to investigate its impact on deep learning algorithms to segment major organs in posterior-anterior chest X-rays. Ps-KDE augments image contrast by substituting pixel values based on their normalized frequency across all images. We evaluate our approach on a U-Net architecture with ResNet34 backbone pre-trained on ImageNet. Five separate models are trained to segment the heart, left lung, right lung, left clavicle, and right clavicle.

### Results

The model trained to segment the left lung using ps-KDE achieved a Dice score of 0.780 (SD = 0.13), while that of trained on CLAHE achieved a Dice score of 0.717 (SD = 0.19), $p<0.01$. ps-KDE also appears to be more robust as CLAHE-based models misclassified right lungs in select test images for the left lung model. The algorithm for performing ps-KDE is available at https://github.com/wyc79/ps-KDE.

Society of Radiological Technology (JSRT) database (http://db.jsrt.or.jp/eng.php).

**Funding:** The author(s) received no specific funding for this work.

## Discussion

Our results suggest that ps-KDE offers advantages over current preprocessing techniques when segmenting certain lung regions. This could be beneficial in subsequent analyses such as disease classification and risk stratification.

## Introduction

### Background

With recent advances in artificial intelligence, deep learning (DL) has emerged as a leading machine-learning technique in medical imaging analysis, playing a transformative role in tasks such as image segmentation [1–3]. This capability extends to various applications, including the segmentation of breast lesions [4, 5], classification of pulmonary cancer stages [6], tissue characterization [7], detection of cardiomegaly [8], and many more. The improved performance for these intricate tasks suggests the potential of computer-aided techniques to improve diagnosis via segmentation.

Within radiology, the segmentation of organs and tumors in medical images holds promise for disease diagnosis and treatment [7]. One approach was the fully convolutional network (FCN), pioneering pixel-to-pixel semantic segmentation [9]. FCN's innovation lies in replacing the last fully connected layer with a deconvolutional layer. Building upon this foundation, the U-Net model as a modification of FCN increases the number of deconvolutional layers and therefore effectively captures more context while requiring smaller training samples [10]. Notably, U-Net has found a widespread application in segmenting medical images across various modalities, including X-rays, Magnetic Resonance Imaging (MRI), Computed Tomography (CT), and histopathology [3, 11, 12].

Current research has focused much on the development of pipelines for the automatic segmentation of medical images, leveraging both preprocessing techniques and the U-Net architecture. A popular generalizable segmentation tool, nnU-Net which was ranked 1st place in Medical Segmentation Decathlon, demonstrated the importance of preprocessing to model performance [13]. Remarkably, even the other simpler U-Net architectures with self-configured preprocessing procedures outperformed more intricate model architectures [14]. Contrast enhancement (i.e., enhancing brightness difference between objects and backgrounds), as a pivotal step for X-ray and CT preprocessing, plays a crucial role in providing human viewers and computer-aided algorithms with crucial features to facilitate analysis [15].

### Related works in contrast enhancement

**Histogram equalization.** Histogram equalization (HE) is a widely used digital image processing method to enhance the contrast of images. It expands an image's distribution range, as some images might only occupy a small portion of the entire value range. The resulting distribution of the pixel value would become more similar to a uniform distribution. However, since most images usually use the whole range of intensity (for instance, 0–255 for a standard RGB image), the HE method would not have much impact on those images [16].

**Adaptive Histogram Equalization (AHE).** The AHE method, as a result, was developed to address this limitation [16]. In AHE, images are divided into subsections, and each subsection is equalized separately. Compared to regular HE, AHE enhances local contrast but with the risk of over-amplifying noise in some regions. Nonetheless, AHE emerged as a popular image preprocessing method in medical imaging applications [15, 17, 18].

**Contrast Limited Adaptive Historgam Equalization (CLAHE).**   An enhancement upon traditional AHE methods, CLAHE, was introduced by clipping histograms to constrain the contrast [16]. It clips the outliers in histograms and redistributes the values across the value range [16]. Recently, several studies have shown the advantages of CLAHE on DL-related tasks, such as predicting five stages of diabetic retinopathy [19], segmenting temporomandibular joint articular disks from MRI [20], and classification of COVID-19 and other pneumonia cases [21].

**Deep-Learning-Based contrast enhancement.**   Neural-network-based image enhancement has emerged in recent years. Anand et al. introduced a contrast diffusion model that learned different contrast levels from low- and high-contrast CXR images [22]. Wei et al. proposed an unsupervised, deep Retinex model for low-light image enhancement via a Decom-Net for decomposition and an Enhance-Net for illumination adjustment [23].

While current methods do exist, there remains a need to pioneer more efficient contrast-enhancement techniques with adequate interpretability. Furthermore, the scarcity of large datasets in real clinical environments poses a challenge to the development of deep-learning-based contrast-enhancement methods that can be generalized effectively.

In this study, we propose a novel, histogram-based, contrast-enhancing method termed ps-KDE. We apply this contrast-enhancement method along with deep-learning segmentation algorithms (e.g., U-Net with ResNet backbones) to the various anatomic structures in a small dataset of X-ray images. We then assess its impact on the performance of deep-learning segmentation based on multiple commonly used evaluation metrics, including Dice/F1-score, Intersection of Union (IoU), recall, and precision. ps-KDE brings three notable contributions: 1) it presents an end-to-end data enhancement method characterized by its simplicity of implementation and adaptability for fine-tuning to accommodate diverse datasets; 2) it demonstrates the efficacy of a density-based augmentation method in segmenting vital organs in chest X-rays; and 3) it establishes the robustness of segmentation algorithms through the interpretation of heatmaps generated by the model.

## Materials and methods

We employed an openly accessible dataset of chest radiographs for our study. The dataset was split equally into a training and a testing set. The images in the training set were augmented through randomized data augmentation and resized to the same resolution for preparation. Then, we performed hyperparameter optimization to identify the optimal parameter configuration. These parameters were integral to the training of our models. After the training phase, the models' performance was evaluated using the test set. S1 Fig illustrates the project's overview in a flowchart format.

### Data

We used a publicly available dataset with 247 posterior-anterior (PA) chest radiographs collected from 13 institutions in Japan and one in the United States. The original radiographs are provided by the Japanese Society of Radiological Technology (JSRT) Database [24] and the manual mask annotations are provided by the Segmentation in Chest Radiology (SCR) Database [25]. The chest radiographs are in PNG format, and the labels are in the form of binary masks. Each image in the database was scanned from film to a size of 2048*2048. Among the 247 images, 154 of them showed solitary pulmonary lung nodules, while the remaining 93 images exhibited no signs of lung nodules. The ethnic representation is unknown.

Among the subset of patients with nodules, gender distribution was observed as 68 males and 86 females. In contrast, among patients without nodules, the gender distribution consisted of 51 males and 42 females. The mean age for patients with nodules is 60 years old. Each image

has five matching masks generated manually by expert radiologists. Each binary mask delineates the boundary of one of the five anatomical structures: heart, left lung, right lung, left clavicle, and right clavicle. Since the original images are in grayscale, with only one color channel, we replicate this single channel to create three channels. This adjustment is necessary to meet the requirement of our deep learning model, which expects inputs to have three color channels.

This study utilizes exclusively publicly available data and thus does not require the Institutional Review Board (IRB) review per regulations set by the Office for Human Research Protections (OHRP) within the U.S. Department of Health and Human Services. The data was accessed on the third day of April 2022. The authors had no access to information that could identify individual participants during or after data collection.

## Data augmentation

Large quantities of data are often needed to train most deep-learning algorithms successfully. Data augmentation is crucial when large datasets are not feasible in order to prevent overfitting and increase model performance. Five types of augmentation were simultaneously applied to each of the original images and its corresponding mask so that the masks correctly represent the anatomical structures on the augmented images. Augmentations include rotation, horizontal flip, vertical flip, a range for image zooms, and rescale. The rotation can occur between 90 degrees clockwise and counterclockwise of the original orientation. Horizontal and vertical flips occur at a probability of 0.5. The range of zoom is between 0.5 and 1.5 for the original images. All images are then rescaled from the red-green-blue scale [0, 255] to [0,1] and resized to 256x256 pixels to help the predictive models achieve faster convergence and higher stability. Fig 1 contains examples of augmentation. Data augmentation was implemented with *ImageDataGenerator* from *TensorFlow* [26].

## Train/Validation split

We randomly split our dataset into training and validation sets. The training set contains 50% of the radiographs (n = 124), and the validation contains the other 50% (n = 123). Having executed five distinct augmentation techniques individually, we expanded our initial training set

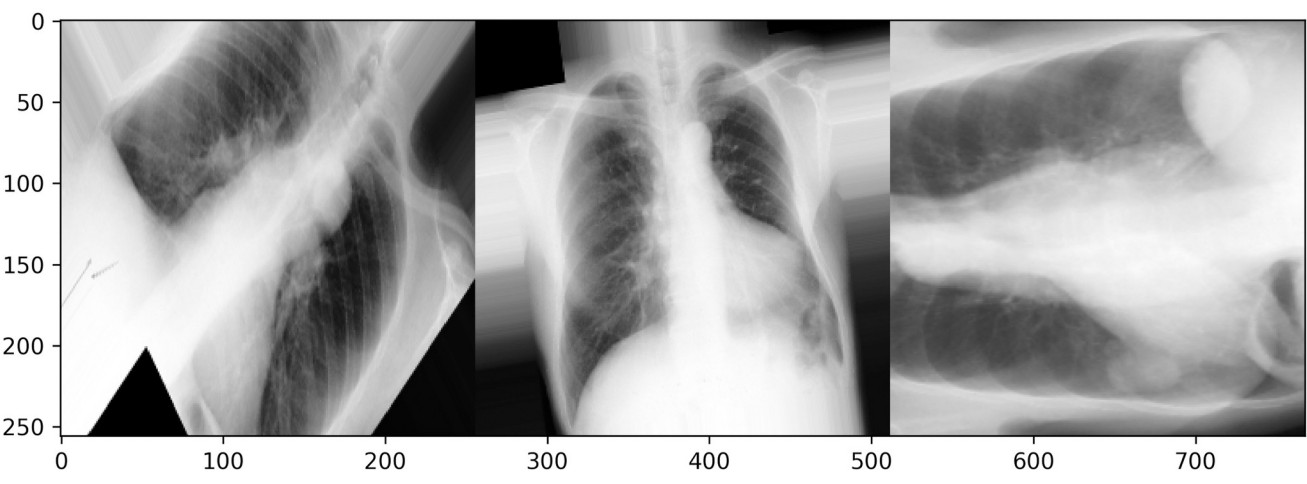

**Fig 1. Augmented images of three distinct individuals in the training set.**

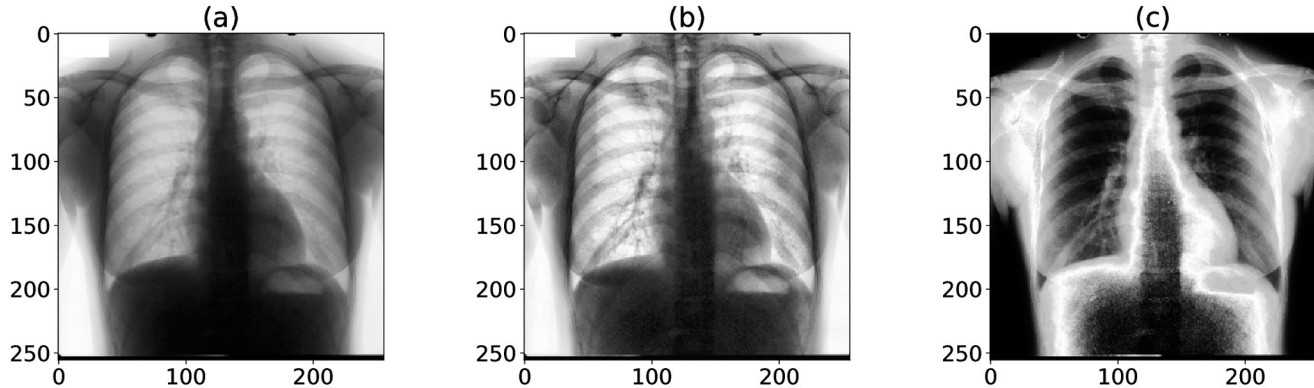

**Fig 2. An example of an X-ray image being processed.** (a) Original chest X-ray image. (b) CLAHE processed chest X-ray image. (c) ps-KDE processed chest X-ray image (location: heart). CLAHE: Contrast Limited Adaptive Histogram Equalization.

to five times its original size, leading to an allocation of 80% for training (n = 620) and 20% for validation (n = 123).

## Image preprocessing

**Contrast Limited Adaptive Histogram Equalization (CLAHE).** We applied CLAHE to our data. An example of chest X-rays preprocessing with CLAHE is shown in Fig 2a and 2b. The equalization of histograms can be visualized in Fig 3a and 3b. The distribution of pixel values became more uniform after CLAHE.

**Pixel-wise substitution by Kernel Density Estimation (ps-KDE).** During the initial exploration of the data, we observed that the distribution of pixel values appeared to be different from organ to organ. We generated histograms of pixel values in different organs to validate our initial observation. We then performed kernel density estimation (KDE) to get a probability density function (PDF) for each organ (Figs 3c and 4a). The PDFs were calculated based on the training set and were stored as prior knowledge. For each image, we substitute each pixel with the density of that pixel value (Fig 4b). The image would then be mapped to a 0–1 range to ensure consistency among images. In other words, our proposed ps-KDE substitutes pixel value for frequency, so that more frequently occurring pixel values in an organ would have a higher value in the resulting plot. Similar to CLAHE, the results were visually appealing (Fig 2c).

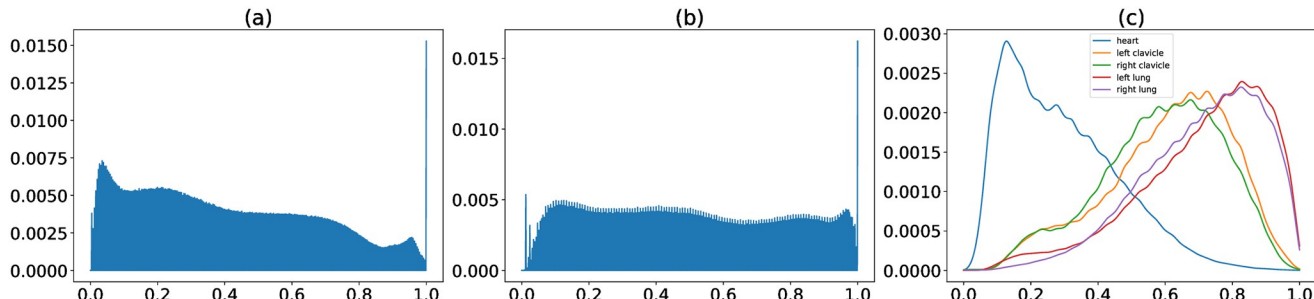

**Fig 3. Normalized distribution of pixel values in X-ray images.** (a) Histogram of original images. (b) Histogram of CLAHE-processed images. (c) KDE of pixel values in different organs. CLAHE: Contrast Limited Adaptive Histogram Equalization; KDE: Kernel density estimation.

(a)    (b)

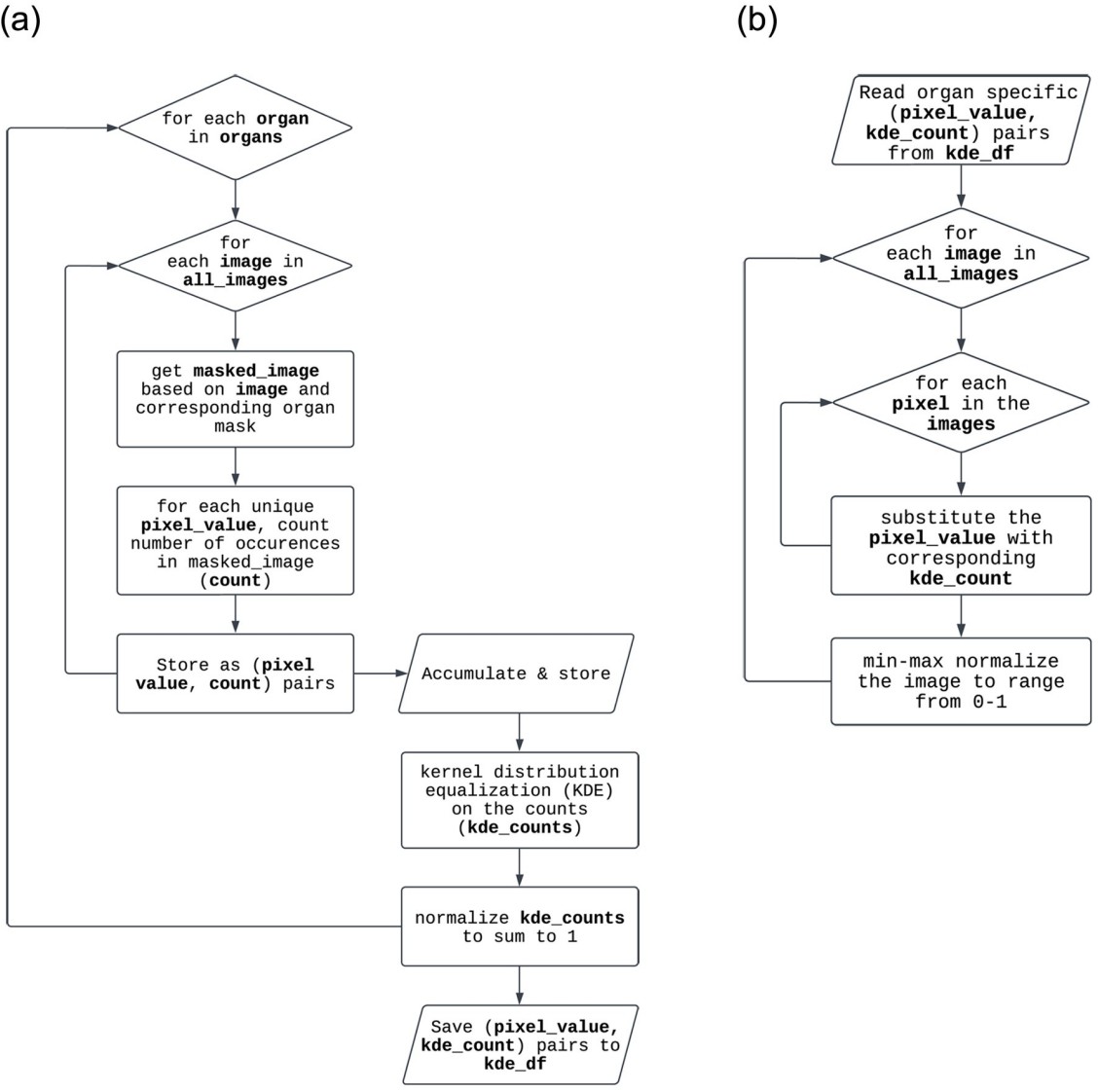

**Fig 4. Flowchart for performing ps-KDE on images.** Codes for performing ps-KDE is available at https://github.com/wyc79/ps-KDE. (a) Algorithm for calculating density using kernel density estimation based on the training set. (b) Algorithm for substituting pixel value with density.

## Model development

We employed a deep learning method for the semantic segmentation of chest radiographs, leveraging the *U-Net* neural network with ResNet backbone designed for segmentation tasks [10].

**Network architecture and implementation.** The network architecture consists of a contracting path and an expansive path. The original design for the contracting path consists of unpadded convolutions with size 3x3, followed by rectified linear units with a 2x2 max-pooling layer, whereas the expansive path applies upsampling for each feature map from the contracting path to restore the original input size. The final layer maps the feature vector to the number of classes.

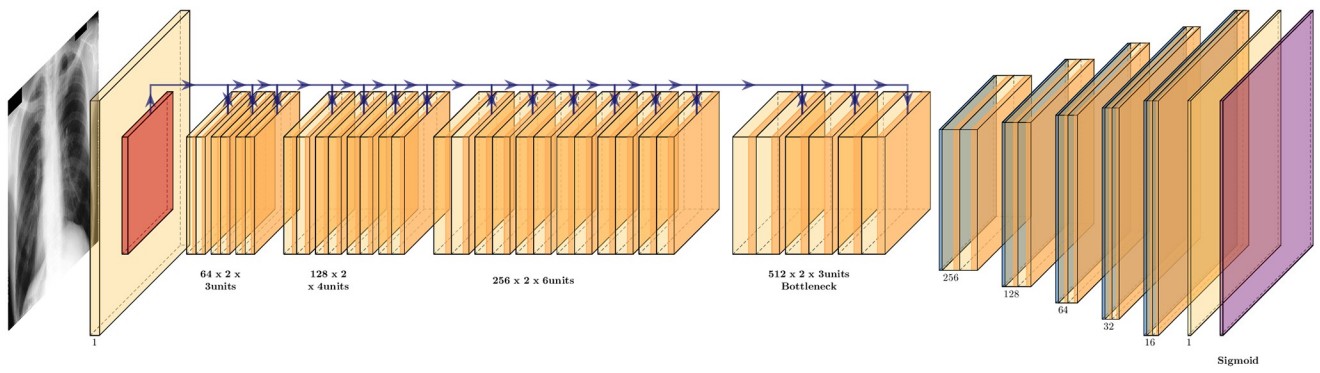

**Fig 5. *ResNetUnet* model.** *U-Net* Model Architecture Implemented with *ResNet34* Backbone.

Implementation-wise, we used the Python package *segmentation_models* (Yakubovskiy, 2019, v1.0.1) with a *ResNet34* backbone featuring pre-trained weights from *ImageNet* (Fig 5). *ResNet* won the *ImageNet* Large Scale Visual Recognition Challenge 2015 with a top-five test error of 3.567 percent in the image classification category [27]. With a network depth of 152, *ResNet* surpasses *VGGNet* in depth by eight times [28]. Referred to as the *ResNetUnet* model in our paper, this amalgamation of *U-Net* and *ResNet34* structures incorporates additional enhancements such as batch normalization and zero padding to complement the original design.

**Loss functions.** A loss function is needed for machine learning models to learn through propagation. Multiple loss functions could be used for image segmentation tasks. For example, three loss functions were proposed to have good performances: binary cross entropy (BCE), binary cross entropy with Jaccard loss (BCE+JCD), and Dice loss (DL). BCE is one of the most commonly used loss functions for machine learning in binary classification tasks. For current work, the mask of each location is a zero or one matrix, which makes the task similar to a pixel-wise binary class classification. Therefore, BCE would be an appropriate loss function to use. The formula for BCE is shown below, considering the ground truth mask *gt* and the model predicted mask *pr*:

$$BCE(gt,\ pr) = -gt \cdot log(pr) - (1 - gt) \cdot log(1 - pr) \tag{1}$$

Another widely used loss function in segmentation tasks is the numeric sum of binary cross entropy and IoU score (Jaccard loss).

$$BCE + JCD(gt, pr) = BCE(gt, pr) + IoU(gt, pr) \tag{2}$$

The Dice coefficient (DC) is a commonly used metric to calculate similarities between images. The Dice coefficient is defined similarly as IoU:

$$DC(gt, pr) = BCE(gt, pr) + IoU(gt, pr) \tag{3}$$

**Model training and hyperparameter optimization.** For optimizing the hyperparameters, we used five-fold cross-validation with all possible combinations of hyperparameters, including the optimizer, loss function, batch size, and learning rate. The list of tuning spaces for each hyper-parameter is shown in Table 1. To search through the proposed space of hyper-parameters, we used a Bayesian optimization process through the *scikit-optimize* package. We first

**Table 1. Search space for hyper-parameters.**

| Hyper-parameters | Tuning Space |
|---|---|
| Optimizer | RMSprop, Adam, SGD |
| Loss Function | BCE, BCE+JCD, DL |
| Batch Size | Integer [1, 6] |
| Learning Rate | Real $[1 \times 10^{-5}]$ |

The categorical and integer variables (optimizer, loss function, and batch size) were initialized to have a uniform prior probability; the learning rate was initialized to have uniform prior distribution in log space (log-uniform). *RMSprop*: *root mean squared propagation; Adam*: *adaptive Moment Estimation; SGD*: *stochastic gradient descent; BCE*: *binary cross entropy; BCE+JCD*: *binary cross entropy with Jaccard loss; DL*: *Dice loss.*

defined an objective function that took instances of hyper-parameters, trained the model, and returned the cross-validation scores (CV scores). We then passed the scores to the optimization function of the package. The optimization process assumed the objective function results to follow a multivariate Gaussian distribution. It would take all observed scores until the current iteration, calculate a posterior distribution and sample the next set of hyper-parameters instances out of the posterior distribution. The best combination of hyper-parameters is chosen for the final model training.

After obtaining the optimized hyperparameters, we fitted models using the original images and two distinct pre-processing techniques (i.e. CLAHE, and ps-KDE) onto the five anatomic structures (i.e., heart, left lung, right lung, left clavicle, right clavicle) with the corresponding best-performing hyperparameters for that task, for a total of 15 models. Our predictive models were then trained with 50 epochs, with $\frac{\# \ of \ training \ sample*2}{batch\_size}$ samples in each step.

## Model evaluation and interpretability

**Evaluation metrics.** We used intersection over union (IoU) and the Dice coefficient (i.e., F-score, Dice score) to evaluate our models. IoU, also known as Jaccard loss, is a commonly used metric in image segmentation tasks. Consider the ground truth mask *gt* and the model predicted mask *pr*:

$$IoU(gt, pr) = \frac{Area(gt \cap pr)}{Area(gt \cup pr)} \qquad (4)$$

We assume that both masks are image matrices of 0's and 1's. Therefore, the area of the mask would be a count of 1's in the corresponding pixel matrix. A high IoU score indicates that more pixels are predicted correctly (more true positives) while fewer pixels are missed (less false negatives and false positives).

F-score, on the other hand, represents a weighted average between precision and recall. In this study, we will report the F1/Dice score. Specifically,

$$F(precision, recall, b = 1) = 2 * \frac{precision * recall}{precision + recall} \qquad (5)$$

We evaluated models on the validation set and reported the mean and standard deviation for each evaluation metric. We performed the independent samples t-test assuming no equal variance to compare the distributions of the Dice scoring metrics between two preprocessing methods using R (version 4.2.3). The significance level (p = 0.01) was not corrected for

multiple comparisons as none of the comparisons was tested more than once. No significance test was performed on precision, recall, IoU, or accuracy.

**Generation of constrast-enhaned images and probability heatmaps.** To understand our models' classification, we randomly chose subjects and obtained the probability of each pixel being classified into the organs or clavicle. A heatmap was produced based on the probabilities using *Matplotlib*. In addition, we overlapped the model's prediction with the original chest X-ray image to evaluate whether the segmentation has clinical merits.

## Result

### Model evaluation

Table 2 demonstrates the results of model optimization based on the cross-validation scheme. The best loss function for all five locations was BCE+JCD, which considers pixel-wise information and intersection maximization.

Table 3 illustrates the evaluation results for various anatomic structures utilizing three distinct image processing techniques. In terms of technique-specific model performance, our analysis revealed that when using the original images (i.e., without CLAHE or ps-KDE transformation), the heart demonstrated the highest segmentation performance, whereas the left clavicle exhibited the least favorable performance based on IoU and Dice scores. With CLAHE transformations, the heart model maintained its superior performance, albeit with the right clavicle registering the lowest scores. With ps-KDE transformation, the five models achieved a mean IoU ranging from 0.577 (SD = 0.06) in the right clavicle to 0.927 (SD = 0.05) in the heart, with Dice scores ranging from 0.275 (SD = 0.17) in the right clavicle to 0.926 (SD = 0.070) in the heart (Fig 6). Across all three techniques, it is noteworthy that the best-performing model differed significantly from the worst-performing one (p < $2.2 \times 10^{-16}$) when assessed by the Dice score. Precision and recall closely mirrored the ranking pattern observed in the Dice score, as anticipated. Accuracy is the highest-performing metric for all three image processing techniques.

Since there is no difference in the ranked order of model performances between the Dice score and mean IoU metrics we will exclusively present the Dice score to assess organ-specific model performances. This decision is made as the five models achieved a lower performance compared to that measured by mean IoU, providing a conservative estimate of the effectiveness of ps-KDE. Notably, significant differences in model performance were observed between the regions classified using CLAHE and ps-KDE. Specifically, in the left lung region, CLAHE had a Dice score of 0.717 (SD = 0.19), and ps-KDE had a Dice score of 0.780 (SD = 0.13), *p* = 0.0026 (Table 3). We observed no differences between the two datasets in heart

**Table 2. Optimization results.**

| Region | Optimizer | Loss Function | Batch Size | Learning Rate | CV Score |
|---|---|---|---|---|---|
| Heart | SGD | BCE+JCD | 1 | 0.07879 | 0.63121 |
| Left Clavicle | SGD | BCE+JCD | 1 | 0.10000 | 0.33311 |
| Left Lung | SGD | BCE+JCD | 1 | 0.00358 | 0.65930 |
| Right Clavicle | RMSprop | BCE+JCD | 1 | 0.00035 | 0.37285 |
| Right Lung | SGD | BCE+JCD | 1 | 0.00139 | 0.73626 |

The best combination for each location is shown in the table. Note that for batch size, the actual batch size used in cross-validation and training models was the above batch size x 5. This multiplier was a result of the data augmentation, as we are loading the original images and augmented images all at the same time.

**Table 3. Model performance after applying preprocessing methods (CLAHE and ps-KDE) evaluated by IoU and Dice scores.**

|  | Region | Original | CLAHE | ps-KDE |
|---|---|---|---|---|
| IoU | Heart | 0.925 (0.07) | 0.921 (0.08) | 0.927 (0.05) |
|  | Left Clavicle | 0.703 (0.12) | 0.778 (0.14) | 0.706 (0.12) |
|  | Left Lung | 0.848 (0.11) | 0.752 (0.12) | 0.799 (0.09) |
|  | Right Clavicle | 0.716 (0.09) | 0.666 (0.14) | 0.577 (0.06) |
|  | Right Lung | 0.863 (0.10) | 0.885 (0.05) | 0.848 (0.09) |
| Dice Score | Heart | 0.918 (0.13) | 0.911 (0.14) | 0.926 (0.07) |
|  | Left Clavicle | 0.537 (0.28) | 0.667 (0.30)* | 0.545 (0.28)* |
|  | Left Lung | 0.839 (0.16) | **0.717 (0.19)*** | **0.780 (0.13)*** |
|  | Right Clavicle | 0.597 (0.17) | 0.440 (0.33)* | 0.275 (0.17)* |
|  | Right Lung | 0.855 (0.17) | 0.894 (0.06)* | 0.850 (0.12)* |
| Precision | Heart | 0.954 (0.09) | 0.947 (0.13) | 0.963 (0.04) |
|  | Left Clavicle | 0.628 (0.32) | 0.781 (0.29) | 0.571 (0.28) |
|  | Left Lung | 0.855 (0.19) | 0.781 (0.21) | 0.927 (0.11) |
|  | Right Clavicle | 0.479 (0.20) | 0.446 (0.33) | 0.233 (0.15) |
|  | Right Lung | 0.946 (0.13) | 0.903 (0.08) | 0.871 (0.10) |
| Recall | Heart | 0.899 (0.15) | 0.889 (0.16) | 0.901 (0.10) |
|  | Left Clavicle | 0.567 (0.35) | 0.628 (0.32) | 0.613 (0.35) |
|  | Left Lung | 0.862 (0.15) | 0.720 (0.21) | 0.702 (0.17) |
|  | Right Clavicle | 0.885 (0.15) | 0.496 (0.40) | 0.394 (0.28) |
|  | Right Lung | 0.798 (0.18) | 0.895 (0.08) | 0.856 (0.15) |
| Accuracy | Heart | 0.987 (0.01) | 0.986 (0.01) | 0.987 (0.01) |
|  | Left Clavicle | 0.993 (0.00) | 0.996 (0.00) | 0.993 (0.00) |
|  | Left Lung | 0.954 (0.04) | 0.924 (0.05) | 0.949 (0.03) |
|  | Right Clavicle | 0.987 (0.01) | 0.992 (0.00) | 0.982 (0.01) |
|  | Right Lung | 0.959 (0.03) | 0.964 (0.02) | 0.950 (0.03) |

IoU and Dice scores are shown as mean (SD). CLAHE: Contrast Limited Adaptive Histogram Equalization; Ps-KDE: Pixel-wise substitution by Kernel Density Estimation; IoU: Intersection over Union.

*:p<0.01 in model performance when comparing between CLAHE and ps-KDE for each segmentation region pair.

segmentation. CLAHE transformation achieved a significant result than ps-KDE in the left clavicle, right clavicle, and right lung.

## Model interpretation

Examples of model predictions with both processing techniques (CLAHE and ps-KDE) are shown in Fig 7. The probability heatmaps showed a decrease in confidence around the edges of the segmentation object. This is more prevalent in the heart and the left clavicle model. Visually, the overlap of the predicted segmentation from ps-KDE and the original x-ray pinpoints the regions that radiologists typically focus on. The partial misclassification in the right lung from the CLAHE technique is discussed in later sections.

## Discussion

In this study, we proposed a novel method, ps-KDE, to substitute the pixel value based on a normalized histogram distribution. Our investigation focused on evaluating the performance of the *ResNetUnet* architecture in the context of segmentation tasks, specifically applied to 247 chest X-rays with PA projection. We assessed each model's segmentation capabilities across

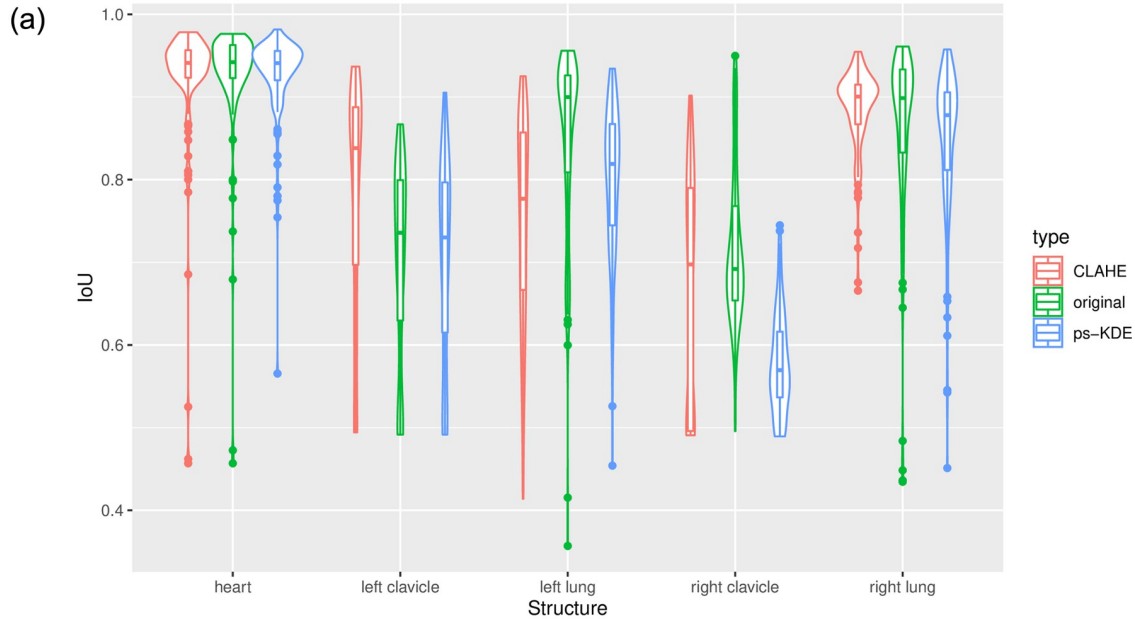

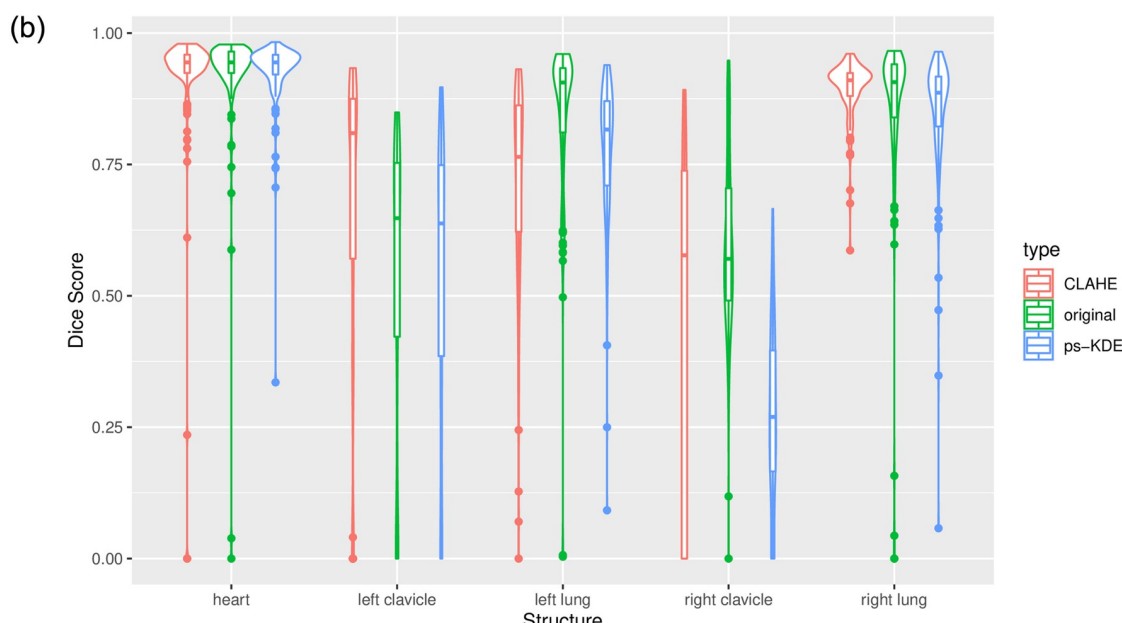

**Fig 6. Model performance represented by violin plots for each anatomical structure.** a) IoU b) Dice score.

five distinct anatomic structures, considering the impact of preprocessing techniques such as ps-KDE and CLAHE. We present ps-KDE as an end-to-end data augmentation method, which transforms raw X-ray images into augmented versions. As an overview, implementing ps-KDE involves traversing a representative image pool to calculate the frequency and density of each pixel value, which is then stored as prior knowledge. Both frequency and density computations can be achieved through a single programming language function call. Subsequently,

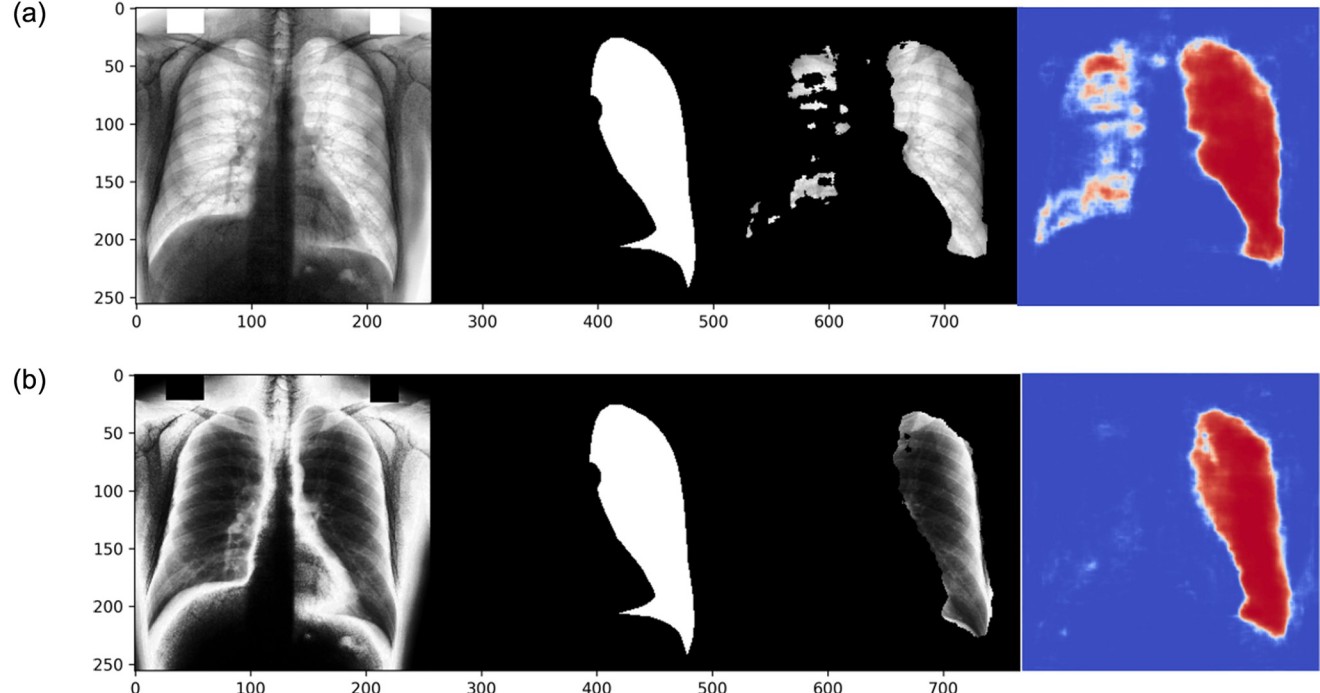

**Fig 7. Prediction of a randomly selected subject.** From left to right, the input of the model, the ground truth, the predicted segmentation overlap with the original x-ray, and the heatmap of the predicted probability. A) CLAHE processed, B) ps-KDE processed.

users only need to assign each pixel value to its corresponding density, making the implementation available to most researchers with minimal programming experience. Crucially, this adaptability allows fine-tuning of ps-KDE to match the representation nuances of various datasets, whether at departmental, institutional, or national scales.

We first compared results within each technique (i.e. original, CLAHE, and ps-KDE), revealing a substantial gap between the highest and lowest Dice scores. These fluctuations should be concerning as test images came from the same dataset. This suggests that although *U-Net* is supposedly designed for end-to-end biomedical image segmentation with very few samples, this algorithm with validation on electronic microscopy image stacks may not generalize to radiographs [10]. We found that, in general, the model predicting lung regions and heart has the highest Dice scores, whereas in the clavicle regions, the Dice Score may drop below 0.6. The higher performance in large regions suggests the model could recognize larger patterns but fell short of smaller ones within the X-Ray.

We then evaluated the efficacy across the three processing techniques on models targeting the same organ. We observed that models preprocessed with CLAHE have higher IoU and Dice scores (Fig 6a and 6b) in the left clavicle regions compared to the original image models. The ps-KDE method, on the other hand, showed better performance in the left lung model than CLAHE. This means the combined use of both preprocessing techniques through a dynamic voting algorithm could be useful by harnessing the advantage of CLAHE in smaller regions and that of ps-KDE in larger regions. The novelty of the ps-KDE method lies within utilizing histogram values not only to generate density estimations but also to execute substitutions. Therefore, such combination allows the pixel substitution to benefit from CLAHE which has a more uniform overall distribution. By enabling accurate and consistent

identification of anatomical structures, our proposed technique stands to enhance the precision of subsequent disease detection algorithms. Furthermore, given its demonstrated superior performance in segmenting specific anatomical structures in chest X-rays, we hypothesize that more advanced imaging techniques, such as CT scans, could potentially benefit from a similar approach.

While ps-KDE showed superior performance in some regions, it is also essential to examine why it may have underperformed in others. Specifically, in the right clavicle region, there is a notable difference in dice score between ps-KDE and CLAHE. Given that the dice score combines precision and recall, examining both metrics reveals that precision shows roughly double the difference compared to recall between ps-KDE and CLAHE. Lower precision indicates a reduced ability to distinguish between true and false positives. Considering that the clavicles are much smaller anatomical structures compared to others, the algorithm might overly contour these regions, leading to reduced precision. This further suggests that ps-KDE may only be suitable for segmenting larger areas. As a potential improvement, it might be worth considering incorporating the density distribution of both the left and right clavicles to double the number of points used for estimating density distribution. Evaluation of performance in the right lung region shows comparable results in terms of precision, recall, and accuracy. However, the reason for the observed superior performance in the left lung but not the right remains unclear. Future investigations could explore how density distributions might adversely affect neural network models [29], potentially guiding improvements to the ps-KDE method by incorporating additional smoothing and density estimation techniques.

We would also like to factor in computational efficiency given its potential impact on the integration of algorithms into clinical practice [30, 31]. CLAHE operates by dividing the input image into tiles and applying histogram equalization to each, thereby obviating the need to modify all pixel values throughout the image [16]. Conversely, ps-KDE relies on a pre-defined frequency table of pixel values, thus necessitating only the referencing of corresponding frequency values for all pixels—a computationally trivial step. While it is worth noting that CLAHE exhibits slightly faster processing times at 0.05(0.01)s and ps-KDE at 0.20(0.11)s, in the context of clinical practice, such discrepancies in turnaround time for radiologists may be negligible. This implies that ps-KDE, alongside other existing contrast enhancement methods offering rapid, on-demand processing speeds, holds promise for seamless integration into existing imaging systems [32], benefiting both clinicians and patients. Moreover, through the establishment of robust and representative pixel frequency tables, this preprocessing method could potentially mitigate systematic biases related to demographics, disease representation, and data management [33, 34].

The incorporation of heatmaps offers invaluable insights into areas of interest and uncertainty during the segmentation process. Notably, we observed a consistent decrease in probability around object edges in the majority of images. This gradual phasing out of probability as the model progresses into negative pixels is ideal, as models exhibiting abrupt switches between high and low confidence levels may lack stability. The visualization of heatmaps also serves to pinpoint regions requiring further investigation. For instance, in CLAHE models, a few misclassifications of the right lung were observed when predicting left lung regions (Fig 7). This may be attributed to image augmentation techniques such as horizontal flips and rotation ranges applied before inputting the images. We hypothesize that, given the small size of our dataset, the spatial distribution of the ground truth significantly influences segmentation outcomes. This suggests that ps-KDE may exhibit greater robustness against substantial image augmentation and small datasets. Future studies could investigate the potential of applying transfer learning to *ResNetUnet* to mitigate the unintended impacts of augmentation [35, 36].

It's worth noting that the predicted lungs still adhere to the clinical expectation that the left lung is narrow and long. Even in cases of misclassification, we can still observe that the model accurately outlines the shape and conforms to the expected characteristics of the right lung. Heatmaps present clinicians with a valuable tool, providing visual assessments of segmentation accuracy and quality. This facilitates interpretation and enables informed clinical decision-making.

## Limitation

Our current dataset contains exclusively PNG images, whereas clinical practices heavily rely on the DICOM format for medical image analysis. While PNG is suitable for research and imaging information in DICOM can be easily converted to PNG format, it lacks the crucial metadata and standardized structure that DICOM would offer. This disconnection hinders the model's direct applicability in clinical settings where DICOM's comprehensive patient information and imaging details are essential.

To mitigate this limitation, the model needs further adaptation for DICOM data format. This involves adjusting the data processing pipeline to handle DICOM images and accounting for metadata intricacies. The model's effectiveness must be re-validated using DICOM data to ensure its reliability in clinical workflows. Addressing this constraint is vital to bridge the gap between research-oriented PNG images and the practical demands of medical professionals who predominantly rely on DICOM for accurate diagnosis and treatment.

We also recognize that the size of our dataset is small for a deep learning algorithm. We also only trained *ResNetUnet* on 50 epochs because of computing resource constraints. Higher performance may be achieved in larger epochs. In addition, the smoothed histogram takes account of only the pixel distribution for this dataset. An additional limitation of our study is the absence of external validation for our models. From a dataset perspective, it remains uncertain how effectively the smoothed histograms can extend to external radiographs, especially those with low quality and contrast. Moreover, there is a potential for another enhanced *U-Net* architecture [37] to provide further validation regarding the applicability of the ps-KDE technique across various model architectures.

## Conclusion

In conclusion, we significantly improved semantic segmentation of the left lung in chest radiographs using ps-KDE. ps-KDE is easy to implement, adaptive across diverse datasets, and enhances the robustness of segmentation algorithms. The introduction of the ps-KDE preprocessing technique contributes to the available image contrasting methods for segmentation but should be treated with caution and further validations.

## Supporting information

**S1 Fig. Overall workflow of the project.**
(PDF)

## Acknowledgments

We thank Kun-Hsing Yu, MD, Ph.D. for assistance with the segmentation algorithm, and Andrew Beam, Ph.D. for comments that greatly improved the experiment design. This work is inspired by the course BMI 707: Deep Learning for Biomedical Data. https://hms-dbmi.github.io/BMI_707/.

## Author Contributions

**Conceptualization:** Yuanchen Wang, Zifan Gu.

**Data curation:** Yujie Guo, Ziqi Wang, Linzi Yu.

**Formal analysis:** Yuanchen Wang, Yujie Yan, Zifan Gu.

**Methodology:** Yuanchen Wang, Zifan Gu.

**Project administration:** Zifan Gu.

**Resources:** Yujie Guo, Ziqi Wang, Linzi Yu.

**Software:** Yujie Yan.

**Visualization:** Yujie Guo, Ziqi Wang, Linzi Yu.

**Writing – original draft:** Yuanchen Wang, Yujie Guo, Ziqi Wang, Linzi Yu, Zifan Gu.

**Writing – review & editing:** Yuanchen Wang, Yujie Yan, Zifan Gu.

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
