## [Decision Letter · Decision Letter 0]

5 Mar 2024

PONE-D-24-05839Enhancing Semantic Segmentation in Chest X-Ray Images through Image Preprocessing: ps-KDE for Pixel-wise Substitution by Kernel Density EstimationPLOS ONE

Dear Dr. Gu,

Thank you for submitting your manuscript to PLOS ONE. After careful consideration, we feel that it has merit but does not fully meet PLOS ONE’s publication criteria as it currently stands. Therefore, we invite you to submit a revised version of the manuscript that addresses the points raised during the review process.

We look forward to receiving your revised manuscript.

Kind regards,

Asadullah Shaikh, Ph.D.

Academic Editor

PLOS ONE

Journal Requirements:

Reviewers' comments:

Reviewer's Responses to Questions

**Comments to the Author**

1. Is the manuscript technically sound, and do the data support the conclusions?

Reviewer #1: Yes

Reviewer #2: Partly

Reviewer #3: Yes

2. Has the statistical analysis been performed appropriately and rigorously? 

Reviewer #1: No

Reviewer #2: I Don't Know

Reviewer #3: Yes

3. Have the authors made all data underlying the findings in their manuscript fully available?

Reviewer #1: Yes

Reviewer #2: Yes

Reviewer #3: Yes

4. Is the manuscript presented in an intelligible fashion and written in standard English?

Reviewer #1: Yes

Reviewer #2: No

Reviewer #3: Yes

5. Review Comments to the Author

Reviewer #1: This is a good manuscript that only needs to undergo a few minor changes:

1-The paper should undergo language editing before it can be published.

2-A literature review section is required.

3-The evaluation should be for the whole model in terms of loss, accuracy, intersection over union (IoU), dice, sensitivity, specificity, specificity, recall, and precision.

4-The pseudocode should be used to write algorithms 1 and 2. Furthermore, a flowchart will be good if included with algorithms.

5-To improve the readability of the paper, I suggest dividing the analysis into several subsections.

6-The authors should explain the dataset split (Training and Testing) rate (90% and 10% or 80% and 20% or 70% and 30%)

7-The author should provide evidence to support that the suggested model is better than the existing models.

8-The discussion and conclusions need to be rewritten to highlight the key contributions of this manuscript.

9-A flowchart for the suggested model is required to show the process steps.

Reviewer #2: The authors submitted a manuscript entitled “Enhancing Semantic Segmentation in Chest X-Ray Images through Image Preprocessing: ps-KDE for Pixel-wise Substitution by Kernel Density Estimation”. The manuscript is well written and have the scope to be published in PLOS ONE, however, it’s current version needs improvement, and there are some concerns that must be addressed in resubmission. The comments are:

1. Define MRI, CT, and other abbreviation in its first place in the document.

2. It would be good to present both of the algorithms in its standard form, not in image or screenshot.

3. There are advance versions of ResNet like ResNet50 etc , why the authors used ResNet34?

4. Result section is short and poor, please improve.

5. In introduction, add more work as the authors did not include literature as a separate section so the introduction part must be longer.

Reviewer #3: The paper hasn’t mentioned the dataset splitting (train, validation, test), this could give the reader in depth knowledge about the experiment.

The paper sorted the limitation while discussing the results, however, for regions where ps-KDE underperformed (such as the Right Clavicle and Right Lung), I recommend providing a more detailed analysis or hypothesis explaining why this might be the case as well as adding more evaluation metrics which help to give wider look in the performance. Including such a discussion would help in understanding the limitations of ps-KDE and in guiding future improvements.

I also recommend expanding the limitation and discussion to discuss the computational efficiency of ps-KDE compared to other methods, especially if the approach is to be recommended for clinical applications where processing speed can be a critical factor.

6. PLOS authors have the option to publish the peer review history of their article (what does this mean?). If published, this will include your full peer review and any attached files.

Reviewer #1: No

Reviewer #2: No

Reviewer #3: No

---

## [Decision Letter · Decision Letter 1]

9 May 2024

Enhancing Semantic Segmentation in Chest X-Ray Images through Image Preprocessing: ps-KDE for Pixel-wise Substitution by Kernel Density Estimation

PONE-D-24-05839R1

Dear Dr. Gu,

We’re pleased to inform you that your manuscript has been judged scientifically suitable for publication and will be formally accepted for publication once it meets all outstanding technical requirements.

Kind regards,

Asadullah Shaikh, Ph.D.

Academic Editor

PLOS ONE

Additional Editor Comments (optional):

Reviewers' comments:

Reviewer's Responses to Questions

**Comments to the Author**

1. If the authors have adequately addressed your comments raised in a previous round of review and you feel that this manuscript is now acceptable for publication, you may indicate that here to bypass the “Comments to the Author” section, enter your conflict of interest statement in the “Confidential to Editor” section, and submit your "Accept" recommendation.

Reviewer #1: All comments have been addressed

Reviewer #3: All comments have been addressed

2. Is the manuscript technically sound, and do the data support the conclusions?

Reviewer #1: Yes

Reviewer #3: Yes

3. Has the statistical analysis been performed appropriately and rigorously? 

Reviewer #1: Yes

Reviewer #3: Yes

4. Have the authors made all data underlying the findings in their manuscript fully available?

Reviewer #1: Yes

Reviewer #3: Yes

5. Is the manuscript presented in an intelligible fashion and written in standard English?

Reviewer #1: Yes

Reviewer #3: Yes

6. Review Comments to the Author

Reviewer #1: Dear Author

Thank you for your response and for taking all the comments in your consideration.

Kind Regards

Reviewer #3: (No Response)

7. PLOS authors have the option to publish the peer review history of their article (what does this mean?). If published, this will include your full peer review and any attached files.

Reviewer #1: No

Reviewer #3: No

---

## [Editor Report · Acceptance letter]

14 May 2024

PONE-D-24-05839R1 

PLOS ONE

Dear Dr. Gu, 

I'm pleased to inform you that your manuscript has been deemed suitable for publication in PLOS ONE. Congratulations! Your manuscript is now being handed over to our production team.

Kind regards, 

on behalf of

Prof. Asadullah Shaikh 

Academic Editor

PLOS ONE